# Near-Data Processing for Machine Learning

**Hyeokjun Choe, Seil Lee, Hyunha Nam, Seongsik Park, Seijoon Kim**
Electrical and Computer Engineering
Seoul National University
Seoul, 08826, Republic of Korea
{genesis1104,lees231,godqhr825,pss015,hokiespa}@snu.ac.kr

**Eui-Young Chung**
Electrical and Electronic Engineering
Yonsei University
Seoul 03722, Republic of Korea
eychung@yonsei.ac.kr

**Sungroh Yoon**[*]
Electrical and Computer Engineering
Seoul National University
Seoul, 08826, Republic of Korea
sryoon@snu.ac.kr

## Abstract

In computer architecture, near-data processing (NDP) refers to augmenting the memory or the storage with processing power so that it can process the data stored therein. By offloading the computational burden of CPU and saving the need for transferring raw data in its entirety, NDP exhibits a great potential for acceleration and power reduction. Despite this potential, specific research activities on NDP have witnessed only limited success until recently, often owing to performance mismatches between logic and memory process technologies that put a limit on the processing capability of memory. Recently, there have been two major changes in the game, igniting the resurgence of NDP with renewed interest. The first is the success of machine learning (ML), which often demands a great deal of computation for training, requiring frequent transfers of big data. The second is the advent of NAND flash-based solid-state drives (SSDs) containing multicore processors that can accommodate extra computation for data processing. Sparked by these application needs and technological support, we evaluate the potential of NDP for ML using a new SSD platform that allows us to simulate in-storage processing (ISP) of ML workloads. Our platform (named ISP-ML) is a full-fledged simulator of a realistic multi-channel SSD that can execute various ML algorithms using the data stored in the SSD. For thorough performance analysis and in-depth comparison with alternatives, we focus on a specific algorithm: stochastic gradient decent (SGD), which is the de facto standard for training differentiable learning machines including deep neural networks. We implement and compare three variants of SGD (synchronous, Downpour, and elastic averaging) using ISP-ML, exploiting the multiple NAND channels for parallelizing SGD. In addition, we compare the performance of ISP and that of conventional in-host processing, revealing the advantages of ISP. Based on the advantages and limitations identified through our experiments, we further discuss directions for future research on ISP for accelerating ML.

## 1 Introduction

Recent successes in deep learning can be accredited to the availability of big data that has made the training of large deep neural networks possible. In the conventional memory hierarchy, the training data stored at the low level (e.g., hard disks) need to be moved upward all the way to the CPU registers. As larger and larger data are being used for training large-scale models such as deep networks (LeCun et al., 2015), the overhead incurred by the data movement in the hierarchy becomes more salient, critically affecting the overall computational efficiency and power consumption.

---

[*]To whom correspondence should be addressed.

The idea of near-data processing (NDP) (Balasubramonian et al., 2014) is to equip the memory or storage with intelligence (i.e., processors) and let it process the data stored therein firsthand. A successful NDP implementation would reduce the data transfers and power consumption, not to mention offloading the computational burden of CPUs. The types of NDP realizations include processing in memory (PIM) (Gokhale et al., 1995) and in-storage processing (ISP) (Acharya et al., 1998; Kim et al., 2016c; Lee et al., 2016; Choi & Kee, 2015). Despite the potential of NDP, it has not been considered significantly for commercial systems. For PIM, there has been a wide performance gap between the separate processes to manufacture logic and memory chips. For ISP, commercial hard disk drives (HDDs), the mainstream storage devices for a long time, normally have limited processing capabilities due to tight selling prices.

Recently, we have seen a resurrection of NDP with renewed interest, which has been triggered by two major factors, one in the application side and the other in the technology side: First, computing- and data-intensive deep learning is rapidly becoming the method of choice for various machine learning tasks. To train deep neural networks, a large volume of data is typically needed to ensure performance. Although GPUs and multicore CPUs often provide an effective means for massive computation required by deep learning, it remains inevitable to store big training data in the storage and then transfer them to the CPU/GPU level for computation. Second, NAND flash-based solid-state drives (SSDs) are becoming popular, gradually replacing HDDs in various computing sectors. To interface SSDs with the host seamlessly replacing HDDs, SSDs require various software running inside, e.g., for address translation and garbage collection (Kim et al., 2002; Gupta et al., 2009). To suit such needs, SSDs are often equipped with multicore processors, which provide far more processing capabilities than those in HDDs. Usually, there exists a plenty of idle time in the processors in SSDs that can be exploited for other purposes than SSD housekeeping (Kim et al., 2010; 2016b).

Motivated by these changes and opportunities, we propose a new SSD platform that allows us to simulate in-storage processing (ISP) of machine learning workloads and evaluate the potential of NDP for machine learning in ISP. Our platform named ISP-ML is a full-fledged system-level simulator of a realistic multi-channel SSD that can execute various machine learning algorithms using the data stored in the SSD. For thorough performance analysis and in-depth comparison with alternatives, we focus on describing our implementation of a specific algorithm in this paper: the stochastic gradient decent (SGD) algorithm, which is the *de facto* standard for training differentiable learning machines including deep neural networks. Specifically, we implement three types of parallel SGD: synchronous SGD (Zinkevich et al., 2010), Downpour SGD (Dean et al., 2012), and elastic averaging SGD (EASGD) (Zhang et al., 2015). We compare the performance of these implementations of parallel SGD using a 10 times amplified version of MNIST (LeCun et al., 1998). Furthermore, to evaluate the effectiveness of ISP-based optimization by SGD, we compare the performance of ISP-based and the conventional in-host processing (IHP)-based optimization.

To the best of the authors' knowledge, this work is one of the first attempts to apply NDP to a multi-channel SSD for accelerating SGD-based optimization for training differentiable learning machines. Our specific contributions can be stated as follows:

- We created a full-fledged ISP-supporting SSD platform called ISP-ML, which required multi-year team efforts. ISP-ML is versatile and can simulate not only storage-related functionalities of a multi-channel SSD but also NDP-related functionalities in realistic manner. ISP-ML can execute various machine learning algorithms using the data stored in the SSD while supporting the simulation of multi-channel NAND flash SSDs to exploit data-level parallelism.

- We thoroughly tested the effectiveness of our platform by implementing and comparing multiple versions of parallel SGD, which is widely used for training various machine learning algorithms including deep learning. We also devised a methodology that can carefully and fairly compare the performance of IHP-based and ISP-based optimization.

- We identified intriguing future research opportunities in terms of exploiting the parallelism provided by the multiple NAND channels inside SSDs. As in high-performance computing, there exist multiple "nodes" (i.e., NAND channel controllers) for sharing workloads, but the communication cost is negligible (due to negligible-latency on-chip communication) unlike the conventional parallel computing. Using our platform, we envision new designs of parallel optimization and training algorithms that can exploit this characteristic, producing enhanced results.

## 2 BACKGROUND AND RELATED WORK

### 2.1 MACHINE LEARNING AS AN OPTIMIZATION PROBLEM

Various types of machine learning algorithms exist (Murphy, 2012; Goodfellow et al., 2016), and their core concept can often be explained using the following equations:

$$F(D, \theta) = L(D, \theta) + r(\theta) \tag{1}$$
$$\theta_{t+1} = \theta_t + \Delta\theta(D) \tag{2}$$
$$\Delta\theta(D) = -\eta \nabla F(D, \theta) \tag{3}$$

where $D$ and $\theta$ denote the input data and model parameters, respectively, and a loss function $L(D, \theta)$ reflects the difference between the optimal and current hypotheses. A regularizer to handle over-fitting is denoted by $r(\theta)$, and the objective function $F(D, \theta)$ is the sum of the loss and regularizer terms. The main purpose of supervised machine learning can then be formulated as finding optimal $\theta$ that minimizes $F(D, \theta)$. Gradient descent is a first-order iterative optimization algorithm to find the minimum value of $F(D, \theta)$ by updating $\theta$ on every iteration $t$ to the direction of negative gradient of $F(D, \theta)$, where $\eta$ is the learning rate. SGD computes the gradient of the parameters and updates them using a single training sample per iteration. Minibatch (stochastic) gradient decent uses multiple (but far less than the whole) samples per iteration. As will be explained shortly, we employ minibatch SGD in our framework, setting the size of a minibatch to the number of training samples in a NAND flash page, which is named 'page-minibatch' (see Figure 2).

### 2.2 PARALLEL AND DISTRIBUTED SGD

Zinkevich et al. (2010) proposed an algorithm that implements parallel SGD in a distributed computing setup. This algorithm often suffers from excessive latency caused by the need for synchronization of all slave nodes. To overcome this weakness, Recht et al. (2011) proposed the lock-free Hogwild! algorithm that can update parameters asynchronously. Hogwild! is normally implemented in a single machine with a multicore processor. Dean et al. (2012) proposed the Downpour SGD for a distributed computing systems by extending the Hodwild! algorithm. While they successfully implemented asynchronous SGD in a distributed computing system, it often fails to overcome communication bottlenecks and shows inefficient bandwidth usage, caused by substantial data movements between computing nodes. Recently proposed EASGD (Zhang et al., 2015) attempted to minimize communication overhead by reducing the frequency of parameter updates. Many EASGD-based approaches reported its effectiveness in distributed environments.

### 2.3 FUNDAMENTALS OF SOLID-STATE DRIVES (SSDS)

SSDs have emerged as a type of next-generation storage device using NAND flash memory (Kim et al., 2010). As shown in the right image in Figure 1(a), a typical SSD consists of an SSD controller, a DRAM buffer, and a NAND flash array. The SSD controller is typically composed of an embedded processor, a cache controller, and channel controllers. The DRAM component, controlled by the cache controller, plays the role of a cache buffer when the NAND flash array is read or written. The NAND flash array contains multiple NAND chips that can be accessed simultaneously thanks to multi-channel configurations and per-channel controllers. Every channel controller is managed by the software called flash translation layer (FTL), which executes wear-leveling and garbage collection to improve the performance and durability of the NAND flash array.

### 2.4 PREVIOUS WORK ON NEAR-DATA PROCESSING

Most of the previous work on ISP focused on popular but inherently simple algorithms, such as scan, join, and query operations (Kim et al., 2016c). Lee et al. (2016) proposed to run the merge operation (frequently used by external sort operation in Hadoop) inside an SSD to reduce IO transfers and read/write operations, also extending the lifetime of the NAND flash inside the SSD. Choi & Kee (2015) implemented algorithms for linear regression, $k$-means, and string match in the flash memory controller (FMC) via reconfigurable stream processors. In addition, they implemented a MapReduce application inside the embedded processor and FMC of the SSD by using partitioning and pipelining methods that could improve performance and reduce power consumption. BlueDBM (Jun

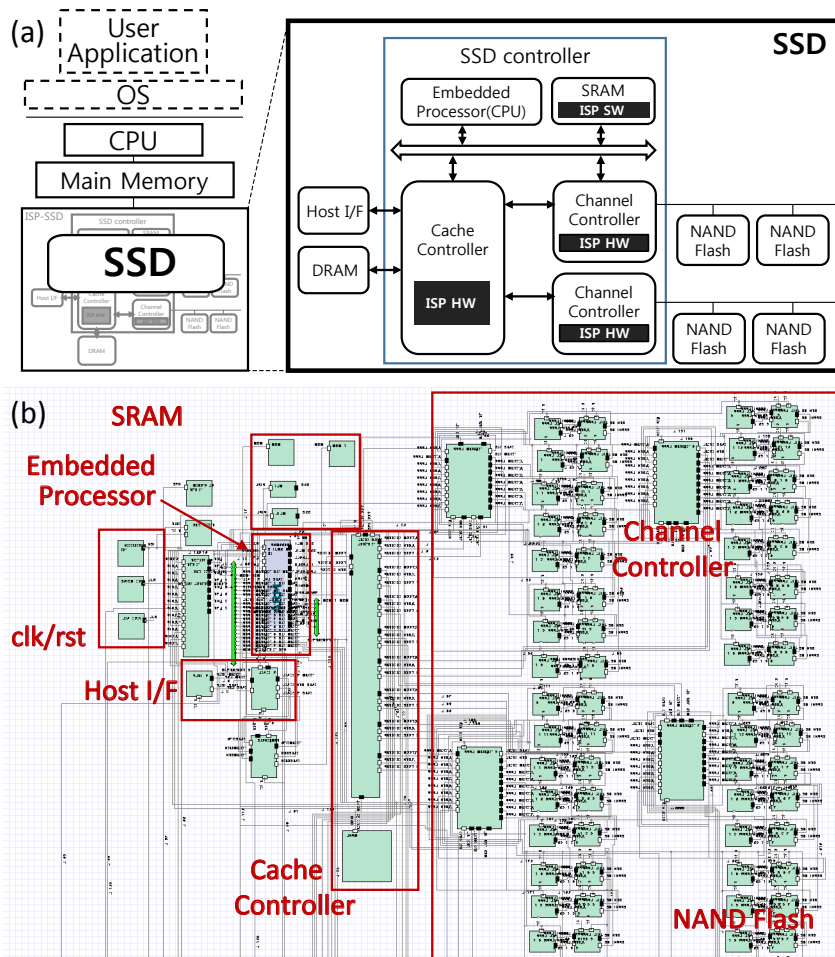

Figure 1: (a) Block diagram of a typical computing system equipped with an SSD and a magnified view of a usual SSD depicting its internal components and their connections. (b) Schematic of the proposed ISP-ML framework, which is implemented in SystemC using Synopsys Platform Architect (http://www.synopsys.com).

et al., 2015) is an ISP system architecture for distributed computing systems with a flash memory-based embedded field programmable gate array (FPGA). The authors implemented nearest-neighbor search, graph traversal, and string search algorithms. No prior work ever implemented and evaluated SSD-based optimization of machine learning algorithms using SGD.

## 3 PROPOSED METHODOLOGY

Figure 1(a) shows the block diagram of a typical computing system, which is assumed to have an SSD as its storage device. Also shown in the figure is a magnified view of the SSD block diagram that shows the major components of an SSD and their interconnections. Starting from the baseline SSD depicted above, we can implement ISP functionalities by modifying the components marked with black boxes (i.e., ISP HW and ISP SW in the figure). Figure 1(b) shows the detailed schematic of our proposed ISP-ML platform that corresponds to the SSD block (with ISP components) shown in Figure 1(a).

In this section, we provide more details of our ISP-ML framework. In addition, we propose a performance comparison methodology that can compare the performance of ISP and the conventional IHP in a fair manner. As a specific example of the ML algorithms that can be implemented in ISP-ML, we utilize parallel SGD.

### 3.1 ISP-ML: ISP Platform for Machine Learning on SSDs

Our ISP-ML is a system-level simulator implemented in SystemC on the Synopsys Platform Architect environment (http://www.synopsys.com). ISP-ML can simulate hardware and software ISP components marked in Figure 1(b) simultaneously. This integrative functionality is crucial for design space exploration in SSD developments. Moreover, ISP-ML allows us to execute various machine learning algorithms described in high-level languages (C or C++) directly on ISP-ML only with minor modifications.

At the conception of this research, we could not find any publicly available SSD simulator that could be modified for implementing ISP functionalities. This motivated us to implement a new simulator. There exist multiple ways of realizing the idea of ISP in an SSD. The first option would be to use the embedded core inside the SSD controller (Figure 1(a)). This option does not require designing a new hardware logic and is also flexible, since the ISP capability is implemented by software. However, this option is not ideal for exploiting hardware acceleration and parallelization. The second option would be to design dedicated hardware logics (such as those boxes with black marks in Figure 1(a) and the entire Figure 1(b)) and integrate them into the SSD controller. Although significantly more efforts are needed for this option compared the first, we chose this second option due to its long-term advantages provided by hardware acceleration and power reduction.

Specifically, we implemented two types of ISP hardware components, in addition to the software components. First, we let each channel controller not only manage read/write operations to/from its NAND flash channel (as in the usual SSDs) but also perform primitive operations on the data stored in its NAND channel. The type of primitive operation performed depends on the machine learning algorithm used (the next subsection explains more details of such operations for SGD). Additionally, each channel controller in ISP-ML (slave) communicates with the cache controller (master) in a master-slave architecture. Second, we designed the cache controller so that it can collect the outcomes from each of the channel controllers, in addition to its inherent functionality as a cache (DRAM) manager inside the SSD controller. This master-slave architecture can be interpreted as a tiny-scale version of the master-slave architecture commonly used in distributed systems. Just as the channel controllers, the exact functionality of the cache controller can be optimized depending on the specific algorithm used. Both the channel controllers and the cache controller have internal memory, but the memory size in the latter is far greater than that in the former.

Specific parameters and considerations used in our implementation can be found in Section 4.1. There are a few points worth mentioning. Unlike existing conventional SSD simulators, the baseline SSD implemented in ISP-ML can store data in the NAND flash memory inside. In order to support reasonable simulation speed, we modeled ISP-ML at cycle-accurate transaction level while minimizing negative impact on accuracy. We omit to describe other minor details of hardware logic implementations, as are beyond the scope of the conference.

### 3.2 Parallel SGD Implementation on ISP-ML

Using our ISP-ML platform, we implemented the three types of parallel SGD algorithms outlined in Figure 2: synchronous SGD (Zinkevich et al., 2010), Downpour SGD (Dean et al., 2012), and EASGD (Zhang et al., 2015). For brevity, we focus on describing the implementation details of these algorithms in ISP-ML and omit the purely algorithmic details of each algorithm; we refer the interested to the corresponding references. Note that the size of a minibatch for the minibatch SGD in our framework is set to the number of training samples in a NAND flash page (referred to as 'page-minibatch' in Figure 2).

For implementing synchronous SGD, we let each of the $n$ channel controllers synchronously compute the gradient. Firstly, each channel controller reads page-sized data from the NAND flash memory and then stores the data in the channel controller's buffer. Secondly, the channel controller pulls the cache controller's parameters ($\theta_{\text{cache}}$) and stores them in the buffer. Using the data and parameters stored in the buffer, each channel controller calculates the gradient in parallel. After transferring the gradient to the cache controller, the channel controllers wait for a signal from the cache controller. The cache controller aggregates and updates the parameters and then sends the channel controller signals to pull and replicate the parameters.

We implemented Downpour SGD in a similar way to implementing synchronous SGD; the major difference is that each channel controller immediately begins the next iteration after transferring the

| **Synchronous SGD** | **Downpour SGD** | **EASGD** |
|---|---|---|
| Processing by *i*-th channel controller and cache controller | Processing by *i*-th channel controller and cache controller | Processing by *i*-th channel controller and cache controller |
| **Repeat**<br> Read a page from NAND<br> pull $\theta_{cache}$<br> $\theta^i = \theta_{cache}$<br> $\Delta\theta^i = 0$<br> **Repeat** for page-minibatch<br> $\theta^i = \theta^i - \eta \, \nabla_t^i(\theta)$<br> $\Delta\theta^i = \Delta\theta^i + \eta \, \nabla_t^i(\theta)$<br> $t{+}{+}$<br> **end**<br> push $\Delta\theta^i$ and wait<br> sync.<br> $\theta_{cache} = \theta_{cache} - 1/n \cdot \Sigma \Delta\theta^i$<br>**end** | **Repeat**<br> Read a page from NAND<br> pull $\theta_{cache}$<br> $\theta^i = \theta_{cache}$<br> $\Delta\theta^i = 0$<br> **Repeat** for page-minibatch<br> $\theta^i = \theta^i - \eta \, \nabla_t^i(\theta)$<br> $\Delta\theta^i = \Delta\theta^i + \eta \, \nabla_t^i(\theta)$<br> $t{+}{+}$<br> **end**<br> **if**($\tau$ devides $t$) **then**<br> push $\Delta\theta^i$<br> $\theta_{cache} = \theta_{cache} - \Delta\theta^i$<br> **end**<br>**end** | **Repeat**<br> Read a page from NAND<br> **Repeat** for page-minibatch<br> $\theta^i = \theta^i - \eta \, \nabla_t^i(\theta)$<br> $t{+}{+}$<br> **end**<br> **if**($\tau$ devides $t$) **then**<br> pull $\theta_{cache}$<br> $\theta^i = \theta - \alpha(\theta^i - \theta_{cache})$<br> push $(\theta^i - \theta_{cache})$<br> $\theta_{cache} = \theta_{cache} + \alpha(\theta^i - \theta_{cache})$<br> **end**<br>**end** |

Figure 2: Pseudo-code of the three SGD algorithms implemented in ISP-ML: synchronous SGD (Zinkevich et al., 2010), Downpour SGD (Dean et al., 2012), and EASGD (Zhang et al., 2015). The shaded line indicates the computation occurring in the cache controller (master); the other lines are executed in the channel controllers (slaves). Note that the term 'page-minibatch' refers to the minibatch SGD used in our framework, where the size of a minibatch is set to the number of training samples in a NAND flash page.

gradient to cache controller. The cache controller updates the parameters with the gradient from the channel controllers sequentially.

For EASGD, we let each of the channel controllers have its own SGD parameters unlike synchronous SGD and Downpour SGD. Each channel controller pulls the parameters from the cache controller after computing the gradient and updating its own parameters. Each channel controller calculates the differences between its own parameters and the cache controller's parameters and then pushes the differences to the cache controller.

Of note is that, besides its widespread use, SGD has some appealing characteristics that facilitate hardware implementations. We can implement parallel SGD on top of the master-slave architecture realized by the cache controller and the channel controllers. We can also take advantage of effective techniques developed in the distributed and parallel computation domain. Importantly, each SGD iteration is so simple that it can be implemented without incurring excessive hardware overhead.

### 3.3 METHODOLOGY FOR IHP-ISP PERFORMANCE COMPARISON

To evaluate the effectiveness of ISP, it is crucial to accurately and fairly compare the performances of ISP and the conventional IHP. However, performing this type of comparison is not trivial (see Section 4.3 for additional discussion). Furthermore, the accurate modeling of commercial SSDs equipped with ISP-ML is impossible due to lack of information about commercial SSDs (e.g., there is no public information on the FTL and internal architectures of any commercial SSD). Therefore, we propose a practical methodology for accurate comparison of IHP and ISP performances, as depicted in Figure 3. Note that this comparison methodology is applicable not only to the parallel SGD implementations explained above but also to other ML algorithms that can be executed in ISP-ML.

In the proposed comparison methodology, we focus on the data IO latency time of the storage (denoted as $T_{\text{IO}}$), since it is the most critical factor among those that affect the execution time of IHP. The total processing time of IHP (IHP$_{\text{time}}$ or $T_{\text{total}}$) can then be divided into the data IO time and the non-data IO time ($T_{\text{nonIO}}$) as follows:

$$\text{IHP}_{\text{time}} = T_{\text{total}} = T_{\text{nonIO}} + T_{\text{IO}}. \tag{4}$$

To calculate the expected IHP simulation time adjusted to ISP-ML, the data IO time of IHP is replaced by the data IO time of the baseline SSD in ISP-ML ($T_{\text{IOsim}}$). By using Eq. (4), the expected IHP simulation time can then be represented by

$$\text{Expected IHP simulation time} = T_{\text{nonIO}} + T_{\text{IOsim}} = T_{\text{total}} - T_{\text{IO}} + T_{\text{IOsim}}. \tag{5}$$

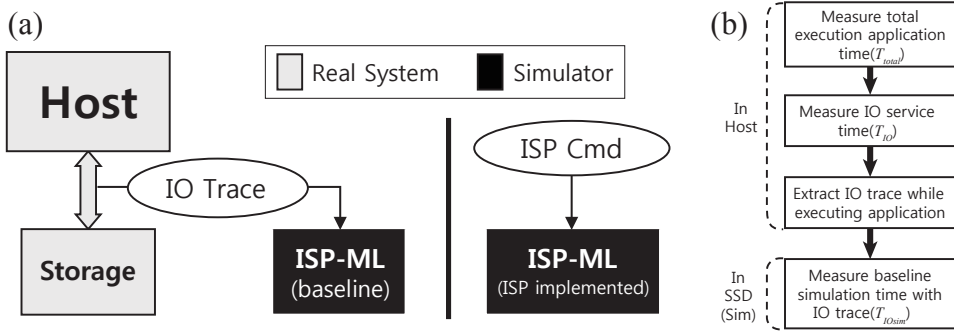

Figure 3: (a) Overview of our methdology to compare the performance of in-host processing (IHP) and in-storage processing (ISP). (b) Details of our IHP-ISP comparison flow.

The overall flow of the proposed comparison methodology is depicted in Figure 3(b). First, the total processing time ($T_{\text{total}}$) and the data IO time of storage ($T_{IO}$) are measured in IHP, extracting the IO trace of storage during an application execution. The simulation IO time ($T_{\text{IOsim}}$) is then measured using the IO trace (extracted from IHP) on the baseline SSD of ISP-ML. Finally, the expected IHP simulation time is calculated by plugging the total processing time ($T_{\text{total}}$), the data IO time of storage ($T_{IO}$) and the simulation IO time ($T_{\text{IOsim}}$) into Eq. (5). With the proposed method and ISP-ML, which is applicable to a variety of IHP environments regardless of the type of storage used, it is possible to quickly and easily compare performances of various ISP implementations and IHP in a simulation environment.

## 4 EXPERIMENTAL RESULTS

### 4.1 SETUP

All the experiments presented in this section were run on a machine equipped with an 8-core Intel(R) Core i7-3770K CPU (3.50GHz) with DDR3 32GB RAM, Samsung SSD 840 Pro, and Ubuntu 14.04 LTS (kernel version: 3.19.0-26-generic). We used ARM 926EJ-S (400MHz) as the embedded processor inside ISP-ML and DFTL (Gupta et al., 2009) as the FTL of ISP-ML. The simulation model we used was derived from a commercial product (Micron NAND MT29F8G08ABACA) and had the following specifications: page size = 8KB, $t_{\text{prog}} = 300\mu$s, $t_{\text{read}} = 75\mu$s, and $t_{\text{block erase}} = 5ms$.[1] Each channel controller had 24KB of memory [8KB (page size) for data and 16KB for ISP] and a floating point unit (FPU) having 0.5 instruction/cycle performance (with pipelining). The cache controller had memory of $(n + 1) \times$ 8KB (page size), where $n$ is the number of channels ($n = 4, 8, 16$). Depending on the algorithm running in ISP-ML, we can adjust these parameters.

Note that the main purpose of our experiments in this paper was to verify the functionality of our ISP-ML framework and to evaluate the effectiveness of ISP over the conventional IHP using SGD, even though our framework is certainly not limited only to SGD. To this end, we selected logistic regression, a fundamental ML algorithm that can directly show the advantage of ISP-based optimizations over IHP-based optimizations without unnecessary complications. We thus implemented the logistic regression algorithm as a single-layer perceptron (with cross entropy loss) in SystemC and uploaded it to ISP-ML. As stated in Section 5.3, our future work includes the implementation and testing of more complicated models (such as deep neural networks) by reflecting the improvement opportunities revealed from the experiments presented in this paper.

As test data, we utilized the samples from the MNIST database (LeCun et al., 1998). To amplify the number of training samples (for showing the scalability of our approach), we used elastic distortion (Simard et al., 2003), producing 10 times more data than the original MNIST (approximately 600,000 training and 10,000 test samples were used in total). To focus on the performance evaluation of running ISP operations, we preloaded our NAND flash simulation model with the simulation

---

[1] These are conservative settings, compared with those of the original commercial product; using the specifications of a commercial product will thus improve the performance of ISP-ML.

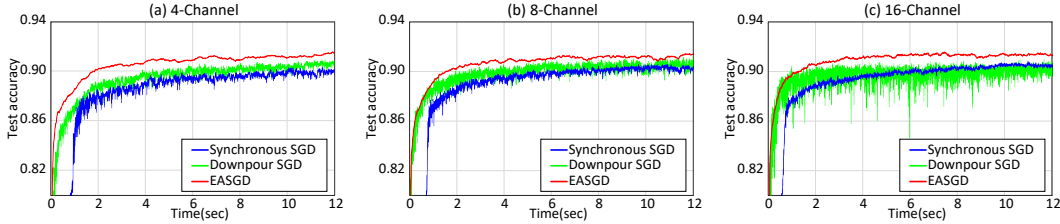

Figure 4: Test accuracy of three ISP-based SGD algorithms versus wall-clock time with a varying number of NAND flash channels: (a) 4 channels, (b) 8 channels, and (c) 16 channels.

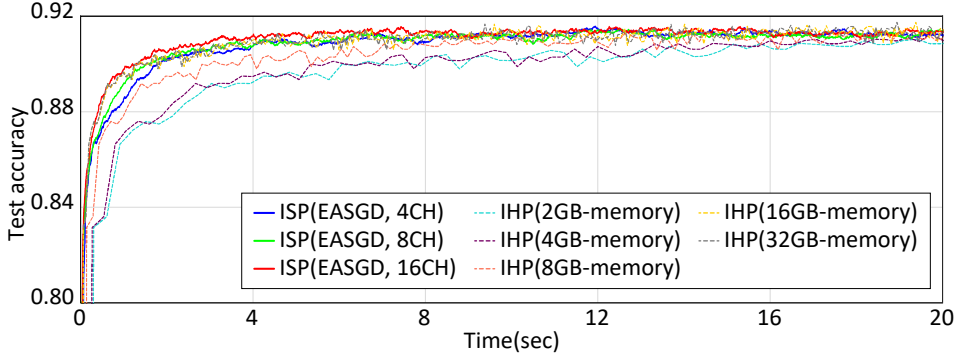

Figure 5: Test accuracy of ISP-based EASGD in the 4, 8, and 16 channel configurations and IHP-based minibatch SGD using diverse memory sizes.

data (the same condition was used for the alternatives for fairness). Based on the size of a training sample this dataset and the size of a NAND page (8KB), we set the size of each minibatch to 10.

## 4.2 PERFORMANCE COMPARISON: ISP-BASED OPTIMIZATION

As previously explained, to identify which SGD algorithm would be best suited for use in ISP, we implemented and analyzed three types of SGD algorithms: synchronous SGD, Downpour SGD, and EASGD. For EASGD, we set the moving rate ($\alpha$) and the communication period ($\tau$) to 0.001 and 1, respectively. For a fair comparison, we chose different learning rates for different algorithms that gave the best performance for each algorithm. Figure 4 shows the test accuracy of three algorithms with varying numbers of channels (4, 8, and 16) with respect to wall-clock time.

As shown in Figure 4, using EASGD gave the best convergence speed in all of the cases tested. EASGD outperformed synchronous and Downpour SGD by factors of 5.24 and 1.96 on average, respectively. Synchronous SGD showed a slower convergence speed when compared to Downpour SGD because it could not start learning on the next set of minibatch until the results of all the channel controllers reported to the cache controller. Moreover, one delayed worker could halt the entire process. This result suggests that EASGD is adequate for all the channel configurations tested in that ISP can benefit from ultra-fast on-chip level communication and employ application-specific hardware that can eliminate any interruptions from other processors.

## 4.3 PERFORMANCE COMPARISON: IHP VERSUS ISP

In large-scale machine learning, the computing systems used may suffer from memory shortage, which incurs significant data swapping overhead. In this regard, ISP can provide an effective solution that can potentially reduce data transfer penalty by processing core operations at the storage level.

In this context, we carried out additional experiments to compare the performance of IHP-based and ISP-based EASGD. We tested the effectiveness of ISP in a memory shortage situation with 5 different configurations of IHP memory: 2GB, 4GB, 8GB, 16GB, and 32GB. We assumed that the host already loaded all of the data to the main memory for IHP. This assumption is realistic because

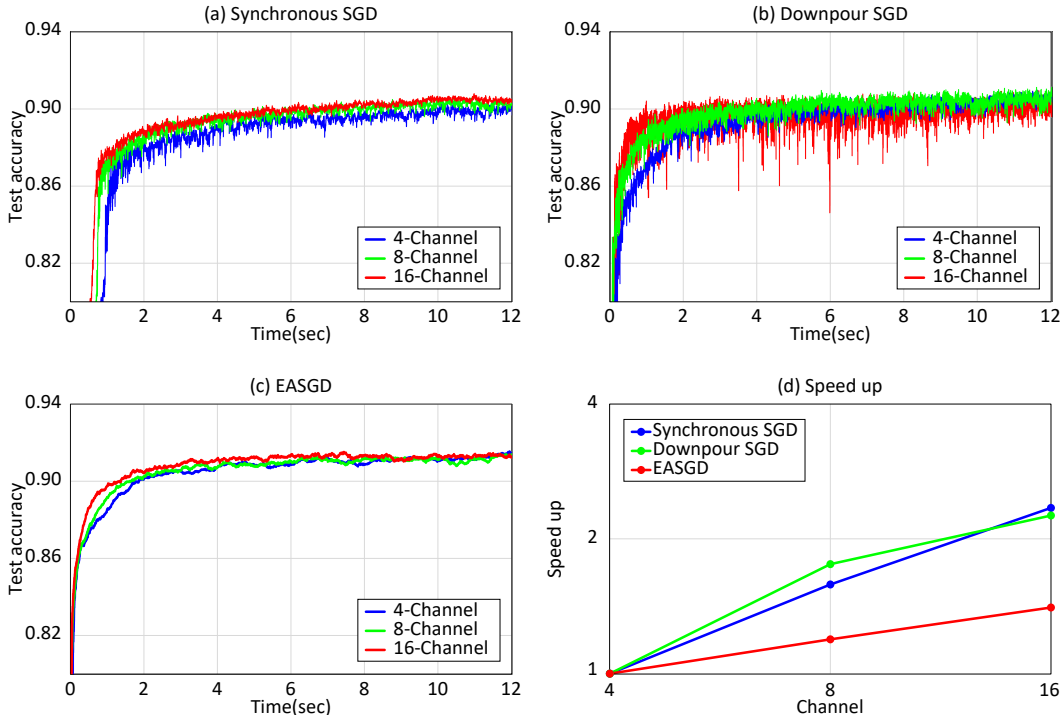

Figure 6: Test accuracy of different ISP-based SGD algorithms for a varied number of channels: (a) synchronous SGD, (b) Downpour SGD, and (c) EASGD. (d) Training speed-up for the three SGD algorithms for a various number of channels.

state-of-the-art machine learning techniques often employ a prefetch strategy to hide the initial data transfer latency.

As depicted in Figure 5, ISP-based EASGD with 16 channels gave the best performance in our experiments. The convergence speed of the IHP-based optimization slowed down, in accordance with the reduced memory size. The results with 16GB and 32GB of memory gave similar results because using 16GB of memory was enough to load and allocate most of the resource required by the process. As a result, ISP was more efficient when memory was insufficient, as would be often the case with large-scale datasets in practice.

## 4.4 CHANNEL PARALLELISM

To closely examine the effect of exploiting data-level parallelism on performance, we compared the accuracy of the three SGD algorithms, varying the number of channels (4, 8, and 16), as shown in Figure 6. All the three algorithms resulted in convergence speed-up by using more channels; synchronous SGD achieved $1.48\times$ speed-up when the number of channels increased from 8 to 16. From Figure 6(d), we can also note that the convergence speed-up tends to be proportional to number of channels. These results suggest that the communication overhead in ISP is negligible, and that ISP does not suffer from the communication bottleneck that commonly occurs in distributed computing systems.

## 4.5 EFFECTS OF COMMUNICATION PERIOD IN ASYNCHRONOUS SGD

Finally, we investigated how changes in the communication period (i.e., how often data exchange occurs during distributed optimization) affect SGD performance in the ISP environment. Figure 7 shows the test accuracy of the Downpour SGD and EASGD algorithms versus wall-clock time when we varied their communication periods. As described in Zhang et al. (2015), Downpour SGD normally achieved a high performance for a low communication period [$\tau = 1, 4$] and became unstable for a high communication period [$\tau = 16, 64$] in ISP. Interestingly, in contrast to the conventional

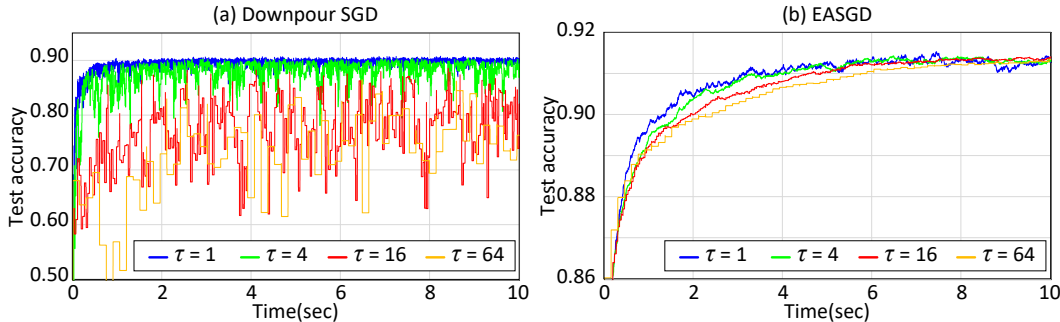

Figure 7: Test accuracy of ISP-based Downpour SGD and EASGD algorithms versus wall-clock time for different communication periods.

distributed computing system setting, the performance of EASGD decreased as the communication period increased in the ISP setting. This is because the on-chip communication overhead in ISP is significantly lower than that in the distributed computing system. As a result, there would be no need for extending the communication period to reduce communication overhead in the ISP environment.

## 5 DISCUSSION

### 5.1 PARALLELISM IN ISP

Given the advances in underlying hardware and semiconductor technology, ISP can provide various advantages for data processing involved in machine learning. For example, our ISP-ML could minimize (practically eliminate) the communication overheads between parallel nodes leveraged by ultra-fast on-chip communication inside an SSD. Minimizing communication overheads can improve various key aspects of data-processing systems, such as energy efficiency, data management, security, and reliability. By exploiting this advantage of fast on-chip communications in ISP, we envision that we will be able to devise a new kind of parallel algorithms for optimization and machine learning running on ISP-based SSDs.

Our experiment results also revealed that a high degree of parallelism could be achieved by increasing the number of channels inside an SSD. Some of the currently available commercial SSDs have as many as 16 channels. Given that the commercial ISP-supporting SSDs would (at least initially) be targeted at high-end SSD markets with many NAND flash channels, our approach is expected to add a valuable functionality to such SSDs. Unless carefully optimized, a conventional distributed system will see diminishing returns as the number of nodes increases, due to the increased communication overhead and other factors. Exploiting a hierarchy of parallelism (i.e., parallel computing nodes, each of which has ISP-based SSDs with parallelism inside) may provide an effective acceleration scheme, although a fair amount of additional research is needed before we can realize this idea.

### 5.2 ISP-IHP COMPARISON METHODOLOGY

To fairly compare the performances of ISP and IHP, it would be ideal to implement ISP-ML in a real semiconductor chip, or to simulate IHP in the ISP-ML framework. Selecting either option, however, is possible but not plausible (at least in academia), because of high cost of manufacturing a chip, and the prohibitively high simulation time for simulating IHP in the Synopsys Platform Architect environment (we would have to implement many components of a modern computer system in order to simulate IHP). Another option would be to implement both ISP and IHP using FPGAs, but it will take another round of significant efforts for developments.

To overcome these challenges (still assuring a fair comparison between ISP and IHP), we have proposed the comparison methodology described in Section 3.3. In terms of measuring the absolute running time, our methodology may not be ideal. However, in terms of highlighting relative performance between alternatives, our method should provide a satisfactory solution.

Our comparison methodology extracts IO trace from the storage while executing an application in the host, which is used for measuring simulation IO time in the baseline SSD in ISP-ML. In this procedure, we assume that the non-IO time of IHP is consistent regardless of the kind of storage the host has. The validity of this assumption is warranted by the fact that the amount of non-IO time changed by the storage is usually negligible compared with the total execution time or IO time.

## 5.3 OPPORTUNITIES FOR FUTURE RESEARCH

In this paper we focused on the implementation and testing of ISP-based SGD as a proof of concept. The simplicity and popularity of (parallel) SGD underlie our choice. By design, it is possible to run other algorithms in our ISP-ML framework immediately; recall that our framework includes a general-purpose ARM processor that can run executables compiled from C/C++ code. However, it would be meaningless just to have an ISP-based implementation, if its performance is unsatisfactory. To unleash the full power of ISP, we need additional ISP-specific optimization efforts, as is typically the case with hardware design.

With this in mind, we have started implementing deep neural networks (with realistic numbers of layers and hyperparameters) using our ISP-ML framework. Especially, we are carefully devising a way of balancing the memory usage in the DRAM buffer, the cache controller, and the channel controllers inside ISP-ML. It would be reasonable to see an SSD with a DRAM cache with a few gigabytes of memory, whereas it is unrealistic to design a channel controller with that much memory. Given that a large amount of memory is needed only to store the parameters of such deep models, and that IHP and ISP have different advantage and disadvantages, it would be intriguing to investigate how to make IHP and ISP can cooperate to enhance the overall performance. For instance, we can let ISP-based SSDs perform low-level data-dependent tasks while assigning high-level tasks to the host, expanding the current roles of the cache controller and the channel controllers inside ISP-ML to the whole system level.

Our future work also includes the following: First, we will be able to implement adaptive optimization algorithms such as Adagrad (Duchi et al., 2011) and Adadelta (Zeiler, 2012). Second, precomputing meta-data during data writes (instead of data reads) could provide another direction of research that can bring even more speedup. Third, we will be able to implement data shuffle functionality in order to maximize the effect of data-level parallelism. Currently, ISP-ML arbitrarily splits the input data into its multi-channel NAND flash array. Fourth, we may investigate the effect of NAND flash design on performance, such as the NAND flash page size. Typically, the size of a NAND flash page significantly affects the performance of SSDs, given that the page size (e.g., 8KB) is the basic unit of NAND operation (read and write). In case where the size of a single example often exceeds the page size, frequent data fragmentation is inevitable, eventually affecting the overall performance. The effectiveness of using multiple page sizes was already reported for conventional SSDs (Kim et al., 2016a), and we may borrow this idea to further optimize ISP-ML.

## ACKNOWLEDGMENTS

The authors would like to thank Byunghan Lee at Data Science Laboratory, Seoul National University for proofreading the manuscript. This work was supported in part by BK21 Plus (Electrical and Computer Engineering, Seoul National University) in 2016, in part by a grant from SK Hynix, and in part by a grant from Samsung Electronics.

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
