# Peer review of "Near-Data Processing for Machine Learning"

_ICLR 2017 — rejected_

[Official Review · AnonReviewer1 · rating 5 · confidence 2 · 16 Dec 2016]
**Near-Data Processing for Machine Learning**

While the idea of moving the processing for machine learning into silicon contained within the (SSD) data storage devices is intriguing and offers the potential for low-power efficient computation, it is a rather specialized topic, so I don't feel it will be of especially wide interest to the ICLR audience. The paper describes simulation results, rather than actual hardware implementation, and describes implementations of existing algorithms. 
The comparisons of algorithms' train/test performance does not seem relevant (since there is no novelty in the algorithms) and the use of a single layer perceptron on MNIST calls into question the practicality of the system, since this is a tiny neural network by today's standards. I did not understand from the paper how it was thought that this could scale to contemporary scaled networks, in terms of numbers of parameters for both storage and bandwidth. 

I am not an expert in this area, so have not evaluated in depth.

[Official Review · AnonReviewer3 · rating 6 · confidence 2 · 17 Dec 2016]
**Interesting topic, hard to fully understand the impact**

Combining storage and processing capabilities is an interesting research topic because data transfer is a major issue for many machine learning tasks.
The paper itself is well-written, but unfortunately addresses a lot of things only to medium depth (probably due length constraints).
My opinion is that a journal with an in-depth discussion of the technical details would be a better target for this paper.

Even though the researchers took an interesting approach to evaluate the performance of the system, it's difficult for me to grasp the expected practical improvements of this approach.
With such a big focus on GPU (and more specialized hardware such as TPUs), the one question that comes to mind: By how much does this - or do you expect it to - beat the latest and greatest GPU on a real task?

I don't consider myself an expert on this topic even though I have some experience with SystemC.

[Official Review · AnonReviewer2 · rating 4 · confidence 4 · 20 Dec 2016]
**Good approach for linear ML, too early for DNNs**

For more than a decade, near data processing has been a key requirement for large scale linear learning platforms, as the time to load the data exceeds the learning time, and this has justified the introduction of approaches such as Spark

Deep learning usually deals with the data that can be contained in a single machine and the bottleneck is often the CPU-GPU bus or the GPU-GPU-bus, so a method that overcomes this bottleneck could be relevant.

Unfortunately, this work is still very preliminary and limited to linear training algorithms, so of little interest yet to ICLR readership. I would recommend publication to a conference where it can reach the large-scale linear ML audience first, such as ICML. This paper is clear and well written in the present form and would probably mostly need a proper benchmark on a large scale linear task. Obviously, when the authors have convincing DNN learning simulations, they are welcome to target ICLR, but can the flash memory FPGA handle it?

For experiments, the choice of MNIST is somewhat bizarre: this task is small and performance is notoriously terrible when using linear approaches (the authors do not even report it)

[Final Decision · Program Chairs · 06 Feb 2017]
**ICLR committee final decision**

This paper is well motivated and clearly written, and is representative of the rapidly growing interdisciplinary area of hardware-software co-design for handling large-scale Machine Learning workloads. In particular, the paper develops a detailed simulator of SSDs with onboard multicore processors so that ML computations can be done near where the data resides.
 
 Reviewers are however unanimously unconvinced about the potential impact of the simulator, and more broadly the relevance to ICLR. The empirical section of the paper is largely focused on benchmarking logistic regression models on MNIST, which reviewers find underwhelming. It is conceivable that the results reflect performance on real hardware, but the ICLR community would atleast expect to see realistic deep learning workloads on larger datasets such as Imagenet, where scalability challenges have been throughly studied. Without such results, the impact of the contribution is hard to evaluate and the claimed gains are bit of a leap of faith. 
 
 The authors make several good points in their response about the paper - that their method is expected to scale, that high quality simulations can given insights that can inform hardware manufacturing, and that their approach complements other hardware and algorithmic acceleration strategies. They are encouraged to resubmit the paper with a stronger empirical section, e.g., benchmarking training and inference of Inception-like models on ImageNet.